# SCHEMA-REFINER: SYNERGIZING KNOWLEDGE GRAPHS AND LLMS FOR PROACTIVE SCHEMA REFINEMENT IN TEXT-TO-SQL

## ABSTRACT

Text-to-SQL broadens data access but often underperforms in real-world databases. We trace a key cause to *lexical-schema ambiguity*, which manifests as homonymy, synonymy, and irregular naming that obscure the mapping between user utterances and schema elements, leading models astray. Prior work primarily deploys interactive, downstream fixes (e.g., robust decoding or clarification), which do not resolve the schemas intrinsic ambiguity. **To mitigate this challenge, we propose Schema-Refiner**, a neuro-symbolic framework that (i) builds a schema knowledge graph, (ii) applies community detection to recover column-level context, (iii) uses a large language model to infer canonical semantics, and (iv) synthesizes `CREATE VIEW` statements, exposing a standardized, disambiguated logical schema layer *for direct consumption by downstream Text-to-SQL models*. This layer leaves the database untouched; a rule-based rewriter maps queries over the views into equivalent SQL on the original schema. To evaluate *robustness to lexicalschema ambiguity* and the effectiveness of our approach, we construct **Amb-Spider** by injecting ambiguities into Spider with human-verified annotations. Across multiple state-of-the-art Text-to-SQL systems, Amb-Spider consistently reduces execution accuracy. When paired with Schema-Refiner, these systems better detect ambiguities and regain a large share of the lost accuracy.

## 1 INTRODUCTION

As a pivotal technology for democratizing data analytics, Text-to-SQL translates natural-language questions-such as "What were the sales trends for top product categories last quarter?"-into executable SQL, enabling non-expert users to access data more easily and strengthening the connection between people and data (Katsogiannis-Meimarakis & Koutrika, 2023). With the rapid progress of Large Language Models (LLMs), the Text-to-SQL task has made substantial advances, and current systems achieve reliable performance on databases with sound data governance (Yu et al., 2018; Zeng et al., 2020; Zhong et al., 2017; Gao et al., 2023). However, in many production environments, inconsistent naming standards, sparse or outdated documentation, and ad hoc schema growth are common, leading to ambiguous or nonstandard table and column names and ultimately degrading Text-to-SQL performance (Fürst et al., 2024; Chang et al., 2023).

These issues encompass ambiguity in the vocabulary used to name tables and columns and typically manifest as (i) homonymy-identical labels for different concepts (e.g., "name" in product and employee tables), (ii) synonymy-different labels for the same concept (e.g., "cust_id" vs. "client_id"), and (iii) irregular naming-nonstandard abbreviations or generic placeholders (e.g., "dept", "val") (Li et al., 2023). For convenience, we refer to this family of phenomena as *lexical–schema ambiguity*. This ambiguity weakens lexical cues for schema linking, increasing table/column mismatches and reducing execution accuracy (Pourreza & Rafiei, 2023).

Although *lexical–schema ambiguity* arises at the schema naming layer, most existing approaches take downstream, reactive routes and thus do not directly disambiguate table/column names (Ding, 2025). At decoding time, systems employ constrained decoding and error correction to improve syntactic validity and handle parsing errors (Wang et al., 2020; Scholak et al., 2021; Chen et al., 2023), which mitigates syntactic failures but leaves naming ambiguity intact. Alternatively, inter-

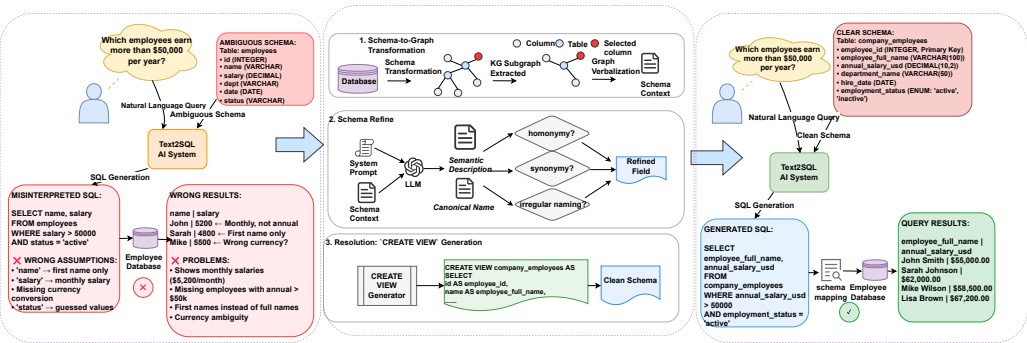

Figure 1: The overall architecture of our Schema-Refiner framework, which diagnoses and resolves schema ambiguity in Text-to-SQL systems.

active mechanisms solicit user feedback for clarification or correction; for example, conversational Text-to-SQL and feedback interfaces allow users to repair parses through natural language or editable step-by-step rationales (Yu et al., 2019; Elgohary et al., 2020; Tian et al., 2023), shifting the burden to users rather than removing ambiguity in the schema. On the evaluation side, ambiguity benchmarks reflect a similar downstream emphasis: AmbiQT and AMBROSIA probe query-level ambiguity or synthetic perturbations rather than schema-internal sources of *lexical–schema ambiguity* (Bhaskar et al., 2023; Saparina & Lapata, 2024). Consequently, these strategies help models cope with the symptoms of ambiguity but largely train them to work around it rather than resolve it at the source, underscoring the need for upstream methods that diagnose and remediate lexical–schema ambiguity before it affects downstream models.

To address this need for upstream solutions, we design the neuro-symbolic method **Schema-Refiner**, illustrated in Figure 1, for automated diagnosis and refinement of database schemas. Our principal contributions are:

1. **An Automated Refinement Method.** We propose **Schema-Refiner**, a neuro-symbolic method that identifies ambiguity hotspots within a schema and synthesizes a disambiguated semantic layer using `CREATE VIEW` statements.

2. **A Diagnostic Benchmark for Evaluation.** To evaluate robustness under *lexical–schema ambiguity*, we construct **Amb-Spider** by injecting homonymy, synonymy, and irregular naming into Spider.

3. **A Systematic Empirical Study.** We conduct a systematic empirical study showing that state-of-the-art Text-to-SQL models suffer degradation on Amb-Spider, and that applying **Schema-Refiner** reduces measured ambiguity and restores execution accuracy across models and databases.

Taken together, these contributions offer an upstream approach for addressing lexical–schema ambiguity that can contribute to improved Text-to-SQL robustness in realistic settings.

## 2 RELATED WORK

**Text-to-SQL.** Text-to-SQL maps natural language questions to executable SQL and has evolved from early neural semantic parsers and cross-domain benchmarks to systems that leverage LLMs (Xinyu et al., 2025; Hong et al., 2025). Core ingredients include schema encoding and linking (Wang et al., 2020), grammar or constraint-based decoding (Scholak et al., 2021), and error correction (Qu et al., 2025; Chaturvedi et al., 2025); recent work further explores LLM-based prompting and tool-augmented NLIDB settings (Wang et al., 2023; Bhaskar et al., 2023). As evaluations move beyond clean benchmarks toward scenarios with noisy schemas and domain-specific conventions, robustness issues become more salient (Renggli et al., 2025). This context motivates a closer look at sources of brittleness rooted in the schema itself.

**Ambiguity Evaluation and Mitigation.** Ambiguity has been examined from two angles: evaluation and mitigation. On the evaluation side, prior work probes ambiguity by allowing multiple gold SQLs or by generating ambiguous questions and schema perturbations, with an emphasis on query-level or synthetic phenomena rather than schema-internal lexical issues (Bhaskar et al., 2023; Saparina & Lapata, 2024; Qiu et al., 2025). On the mitigation side, downstream methods span decoding-time constraints and error handling, prompt-based and self-refinement procedures, and interactive clarification with users (see also the Text-to-SQL discussion), for which overviews are provided by recent surveys (Wang et al., 2023; Xinyu et al., 2025) and representative systems (Yu et al., 2019; Elgohary et al., 2020; Tian et al., 2023; Qiu et al., 2025). These approaches generally treat the schema as fixed and address ambiguity at inference time. Emerging upstream considerations enrich schema context or distill facts, but typically do not provide a formal definition of lexical–schema ambiguity, a model-agnostic metric for quantification, or durable schema-level remediation artifacts (Chen et al., 2025).

## 3 METHODOLOGY

Our approach proceeds from a formalization of lexical–schema ambiguity via a Semantic Similarity Score (SSS) to an automated refinement system. We define ambiguity types over database schemas and present **Schema-Refiner**, a neuro-symbolic framework that diagnoses and resolves ambiguity by synthesizing a non-destructive semantic layer through `CREATE VIEW`. An overview is shown in Figure 1.

### 3.1 PROBLEM SETUP AND NOTATION.

Let a schema $D$ comprise tables $T = \{t_1, \ldots, t_n\}$ and columns $C = \{c_1, \ldots, c_m\}$. Each column $c \in C$ belongs to a table $\texttt{table}(c)$ and has a surface name $\texttt{name}(c)$. We posit abstract *Semantic Concepts* $\mathcal{S} = \{s_1, \ldots, s_k\}$ and an ideal mapping $\Phi : C \to \mathcal{S}$ that assigns each column to its intended concept (a conceptual ground truth used to define ambiguity). Let $\texttt{canonical\_name}(s)$ denote the canonical string for $s \in \mathcal{S}$; canonical names are inferred downstream by our pipeline.

To quantify semantic relatedness, we use a Semantic Similarity Score $\text{SSS}(\cdot, \cdot) \in [0, 1]$. Let $r(\cdot)$ be a representation function that maps either concepts $s \in \mathcal{S}$ or strings (e.g., surface names) to embeddings, and let $\text{sim}(\cdot, \cdot)$ be a similarity function (e.g., cosine). We define

$$\text{SSS}(x, y) = \text{sim}\big(r(x), r(y)\big), \qquad x, y \in \mathcal{S} \cup \text{Strings}, \tag{1}$$

where higher values indicate greater semantic similarity. We use thresholds $\tau_h, \tau_s, \tau_i \in (0, 1)$ for homonymy, synonymy, and irregular naming, respectively.

**Ambiguity Types.** We focus on three lexical–schema phenomena and characterize them with SSS thresholds.

**Homonymy.** Two columns share the same surface name but refer to different concepts, evidenced by low concept-level similarity:

$$\text{Homonymy}(c_i, c_j) \iff \texttt{name}(c_i) = \texttt{name}(c_j) \land \text{SSS}\big(\Phi(c_i), \Phi(c_j)\big) < \tau_h. \tag{2}$$

**Synonymy.** Two columns have different surface names but refer to the same concept, evidenced by high concept-level similarity:

$$\text{Synonymy}(c_i, c_j) \iff \texttt{name}(c_i) \neq \texttt{name}(c_j) \land \text{SSS}\big(\Phi(c_i), \Phi(c_j)\big) \geq \tau_s. \tag{3}$$

**Irregular naming.** A columns surface name deviates from its canonical representation for the intended concept:

$$\text{Irregular}(c) \iff \text{SSS}\big(\texttt{name}(c), \texttt{canonical\_name}(\Phi(c))\big) < \tau_i. \tag{4}$$

### 3.2 PIPELINE OVERVIEW

We adopt a two-stage pipeline: (i) an *inference stage* that derives a semantic package for each column from schema context, and (ii) an *evaluation stage* that detects three ambiguity types using SSS-based rules with explicit precedence.

STAGE I: INFERENCE

1. **Schema to Graph.** Convert the database schema into a heterogeneous knowledge graph $G$ with `Table`/`Column` nodes and `:HAS_COLUMN`/`:FOREIGN_KEY` edges.

2. **Community Context.** Project to a column graph and run Leiden to obtain communities; for each target column $c$, take its community-induced subgraph as its structural context.

3. **Augmented Context Generation and LLM Inference.** To create a rich, multi-faceted context for column $c$, we supplement its structural context with data-level evidence. Specifically, we **actively probe** the database to retrieve a few representative data samples from the column. This data is then serialized along with the community-induced subgraph into a comprehensive textual prompt. Finally, we query an LLM with this prompt to produce the semantic package:

$$\text{INFERSEMANTICPACKAGE}(c) = \big(\text{desc}(c), \text{canon}(c)\big), \tag{5}$$

where $\text{desc}(c)$ is a short semantic description and $\text{canon}(c)$ is the canonical name inferred from context.

STAGE II: EVALUATION

Let $\text{SSS}(\cdot, \cdot) \in [0, 1]$ denote the Semantic Similarity Score, and let $\tau_{\text{self}}$, $\sigma_{\text{sim}}$, $\sigma_{\text{dissim}} \in (0, 1)$ be thresholds.

**Candidate formation (blocking).**

- **Homonymy** candidates: all pairs $(c_i, c_j)$ with identical surface names $\text{name}(c_i) = \text{name}(c_j)$.
- **Synonymy** candidates: all pairs $(c_i, c_j)$ with $\text{name}(c_i) \neq \text{name}(c_j)$ and compatible types (e.g., both numeric/ID/date).
- **Irregular naming**: all single columns $c$.

**SSS-based decisions.**

$$\text{Homonymy}(c_i, c_j) \iff \text{name}(c_i) = \text{name}(c_j) \wedge \text{SSS}\big(\text{desc}(c_i), \text{desc}(c_j)\big) < \sigma_{\text{dissim}}. \tag{6}$$

$$\text{Synonymy}(c_i, c_j) \iff \text{name}(c_i) \neq \text{name}(c_j) \wedge \text{SSS}\big(\text{desc}(c_i), \text{desc}(c_j)\big) \geq \sigma_{\text{sim}}. \tag{7}$$

$$\text{Irregular}(c) \iff \text{SSS}\big(\text{name}(c), \text{canon}(c)\big) < \tau_{\text{self}}. \tag{8}$$

**Precedence rule.** Homonymy and Synonymy take precedence over Irregular naming: once a pair $(c_i, c_j)$ is labeled by Eq. equation 6 or Eq. equation 7, the involved columns are not reconsidered by Eq. equation 8. This avoids double counting and prioritizes inter-column ambiguity over name canonicalization.

**Output and non-destructive use.** We expose a logical view layer by synthesizing `CREATE VIEW` statements with repaired aliases; no physical changes are made to the base schema. Queries issued over the view layer are deterministically rewritten into equivalent SQL over the original tables.

# 4 EXPERIMENTS

## 4.1 RESEARCH QUESTIONS

Our experiments are designed to answer three questions that form a logical narrative, moving from problem validation to solution efficacy:

- **RQ1 (Impact):** Does schema-level lexical ambiguity degrade the performance of state-of-the-art Text-to-SQL systems, even when the underlying database semantics and user questions remain constant?

- **RQ2 (Identification):** How effectively can our proposed SCHEMA-REFINER identify these ambiguities, compared to practical, context-aware baselines?
- **RQ3 (Recovery):** Does applying SCHEMA-REFINER's non-destructive view layer enable these systems to recover the performance lost due to ambiguity?

The following sections detail the experimental setup designed to address these questions, with our results structured to answer each RQ in sequence.

## 4.2 EXPERIMENTAL SETUP

### 4.2.1 DATASETS AND AMBIGUITY INJECTION

To address our research questions, we use two benchmarks: the widely-used **Spider** benchmark and our proposed ambiguous counterpart, **Amb-Spider**.

**Amb-Spider** is constructed on top of Spider to systematically evaluate the impact of schema-level lexical ambiguity (**RQ1**). It introduces three controlled types of ambiguity: *homonymy*, *synonymy*, and *irregular naming*. We generated Amb-Spider via an LLM-driven injection pipeline, followed by manual auditing, to ensure semantic preservation. For every database in Spider's development and test splits, a corresponding schema exists in Amb-Spider, and we use the *same* question sets. This one-to-one correspondence enables a fair, controlled comparison.

### 4.2.2 SYSTEMS UNDER TEST

We evaluate three high-performing and representative Text-to-SQL systems to assess the impact of ambiguity and the potential for recovery (**RQ1** and **RQ3**):

- **DIN-SQL** (Pourreza & Rafiei, 2023), which decomposes the text-to-SQL task into distinct modules for schema linking, query classification, SQL generation, and self-correction.
- **DAIL-SQL** (Gao et al., 2023), wDAIL-SQL (Gao et al., 2023), which proposes a token-efficient prompt strategy using skeleton-based example selection for in-context learning.
- **MAC-SQL** (Wang et al., 2025), which utilizes a multi-agent collaborative framework for systematic problem decomposition and feedback-driven query refinement.

For each system, we use their publicly available code and default configurations without any further fine-tuning.

### 4.2.3 EVALUATION PROTOCOL AND METRICS

We employ different metrics tailored to our research questions:

**For Text-to-SQL Performance (RQ1 and RQ3):** We report **Execution Accuracy (ExAcc)**, which measures the percentage of generated SQL queries that execute correctly and produce the expected result. To ensure robustness, all reported ExAcc scores are the mean of **five independent runs** with different random seeds (starting from 42). When evaluating recovery for **RQ3**, SQL queries generated against the refined schema are deterministically mapped back to the original Amb-Spider schema for execution via a rule-based converter.

**For Ambiguity Identification (RQ2):** We evaluate the performance of our SCHEMA-REFINER using standard classification metrics: **Precision, Recall, and F1-score**. These are calculated for each of the three ambiguity types (homonymy, synonymy, and irregular naming) to assess our method's diagnostic accuracy.

### 4.2.4 IMPLEMENTATION DETAILS

For all automated data generation and ambiguity injection tasks, we use the GPT-4.1 model with a temperature of 0 for reproducibility. For all internal text representation tasks, such as computing semantic similarity, we consistently use the text-embedding-3-large model.

Table 1: RQ1: Execution Accuracy (ExAcc, %) on Spider ("Original") vs. Amb-Spider ("Ambiguous"). The $\Delta$ column shows the performance drop. Results are from deterministic runs with a fixed seed of 42, confirming stable outcomes.

| Model | Dev | | Test | |
|---|---|---|---|---|
| | Spider | Amb-Spider ($\Delta$) | Spider | Amb-Spider ($\Delta$) |
| DIN-SQL | 82.60 | 58.20 (-24.40) | 78.90 | 65.60 (-13.30) |
| DAIL-SQL | 83.00 | 59.10 (-23.90) | 83.20 | 58.40 (-24.80) |
| MAC-SQL | 85.33 | 78.80 (-6.53) | 82.40 | 78.80 (-3.60) |

### 4.3 RQ1: Impact of Lexical Ambiguity on Text-to-SQL Performance

To quantify the impact of schema-level lexical ambiguity, we compare the performance of three leading Text-to-SQL systems on the original **Spider** benchmark against our **Amb-Spider** counterpart, using identical question splits. The results, presented in Table 1, reveal a substantial degradation in execution accuracy across the board.

The performance drop is particularly severe for DIN-SQL and DAIL-SQL, which experience a decline of over 20 percentage points in accuracy on multiple splits, indicating their architectures are highly sensitive to the surface forms of schema names. In contrast, MAC-SQL demonstrates greater resilience with a more contained drop of 3-7 points.

**Key Finding:** Schema-level lexical ambiguity is not a minor nuisance but a critical bottleneck for modern Text-to-SQL systems. The significant performance degradation across all tested models confirms the necessity of an upstream schema refinement stage, which we evaluate in the subsequent sections.

### 4.4 RQ2: Accuracy of Lexical Ambiguity Identification

#### 4.4.1 Experimental Design: Task, Methods, and Metrics

This experiment evaluates whether SCHEMA-REFINER can accurately identify schema-level lexical ambiguity, outperforming practical baselines. The task is a multi-label classification problem: for each column in an Amb-Spider database, the system must predict if it exhibits **homonymy**, **synonymy**, or **irregular naming**.

We compare our method against two strong baseline strategies that represent common, intuitive approaches to context serialization:

- **Table-only**: Provides only the schema of the column's host table as context. This simulates a minimal-context scenario.

- **Full-schema**: Provides the entire database schema as a flat text file. This simulates a maximal, but noisy, context scenario.

In contrast, SCHEMA-REFINER uses an intermediate approach by extracting a semantically relevant community as context via the Leiden algorithm. Both our method and the baselines use the same GPT-4.1 and text-embedding-3-large models to ensure a fair comparison focused solely on the quality of the provided context.

Performance is measured using micro-averaged **Precision**, **Recall**, and **F1-score**.

#### 4.4.2 Fair Threshold Selection

A critical aspect of our method is the use of three similarity thresholds. To ensure a fair evaluation and prevent test set leakage, these thresholds are selected *a priori* on a held-out validation split via a $4 \times 4 \times 4$ grid search that maximizes F1-score. The final selected thresholds, $\tau_{\text{irreg}}=0.60$, $\sigma_{\text{syn}}=0.90$, and $\sigma_{\text{hom}}=0.75$, are used for all experiments without any test-time tuning.

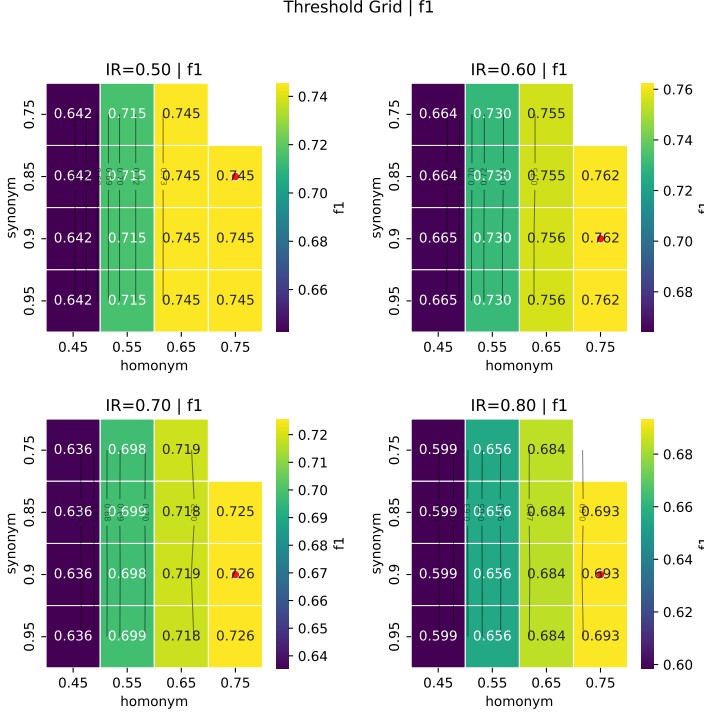

Figure 2: Threshold selection via grid search on a held-out validation split. The star indicates the final configuration chosen for all experiments.

### 4.4.3 MAIN RESULTS AND ABLATION STUDY

As summarized in Table 2, SCHEMA-REFINER achieves the highest micro-averaged F1-score, indicating a superior balance of precision and recall. This confirms that its intelligent, community-derived context is more effective than either the minimal (table-only) or the noisy, unfiltered (full-schema) context.

Table 2: RQ2 (Overall): Micro-averaged ambiguity identification results (%) on Amb-Spider.

| Method | Dev | | | Test | | |
|---|---|---|---|---|---|---|
| | Prec. | Rec. | F1 | Prec. | Rec. | F1 |
| Table-only | 71.2 | 79.3 | 75.2 | 75.3 | 72.8 | 74.0 |
| Full-schema | 59.7 | 72.9 | 65.9 | 72.8 | 73.8 | 73.3 |
| **Schema-Refiner** | **73.0** | **79.7** | **76.2** | **76.4** | **78.9** | **77.7** |

To isolate the contribution of individual components, we conducted an ablation study on the active probing (AP) mechanism. As shown in Table 3, removing this component leads to a notable performance drop, demonstrating its critical role. On the development set, the full model with AP outperforms the variant without it across all metrics, achieving a 3.4-point F1-score improvement.

More revealingly, on the test set, AP yields a substantial 4.4-point gain in recall for a slight cost in precision. This trade-off is highly informative: it suggests that active probing empowers the model to uncover more complex and subtle ambiguities that would otherwise be missed. The resulting net gain in the F1-score (+1.5 points) confirms that this is a worthwhile trade-off and highlights AP as a vital component for maximizing identification coverage.

Table 3: RQ2 (Ablation): Effect of active probing (AP) on ambiguity identification results (%).

| Variant | Dev | | | Test | | |
|---|---|---|---|---|---|---|
| | Prec. | Rec. | F1 | Prec. | Rec. | F1 |
| Schema-Refiner (no AP) | 68.7 | 77.4 | 72.8 | 78.0 | 74.5 | 76.2 |
| **Schema-Refiner (w/ AP)** | **73.0** | **79.7** | **76.2** | **76.4** | **78.9** | **77.7** |

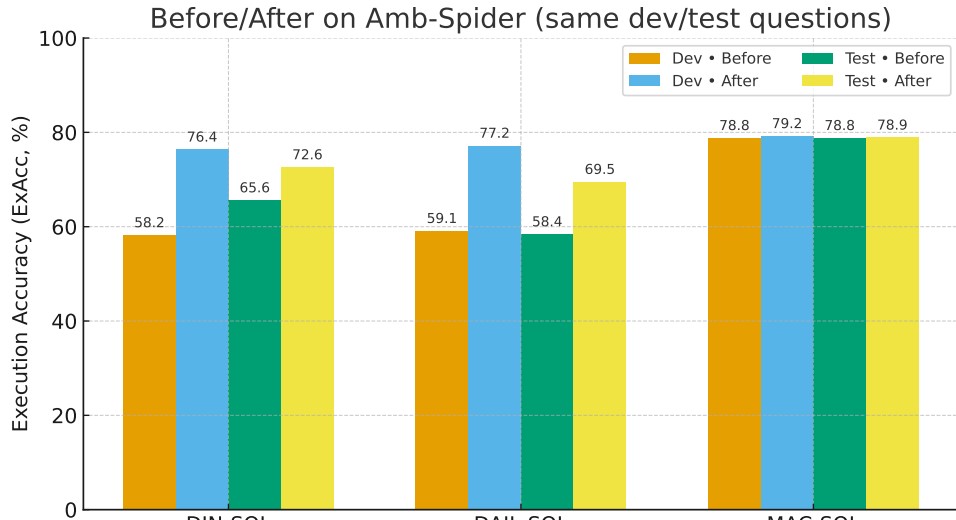

Figure 3: Execution Accuracy (ExAcc, %) on Amb-Spider before and after applying SCHEMA-REFINER. The refinement process provides significant gains for DIN-SQL and DAIL-SQL, which were more sensitive to the initial ambiguity.

## 4.5 RQ3: PERFORMANCE RECOVERY VIA UPSTREAM REFINEMENT

### 4.5.1 EXPERIMENTAL PROTOCOL

To assess whether our refined schema can recover lost accuracy, we apply SCHEMA-REFINER as an upstream pre-processing step. For each database in Amb-Spider, the Text-to-SQL models now interact with the clean, refined schema (materialized as database views). The generated SQL queries, which reference the clean aliases, are then passed through a deterministic, rule-based back-mapping converter. This converter replaces the aliases with their original, ambiguous names, allowing the queries to be executed on the untouched database. This protocol ensures that any performance change is due solely to the disambiguation provided during generation. All other experimental settings are identical to those in RQ1.

### 4.5.2 RESULTS: SUBSTANTIAL RECOVERY AND CEILING EFFECTS

The results, visualized in Figure 3 and quantified in Table 4, demonstrate that schema refinement yields substantial performance recovery for the models most affected by ambiguity.

The efficacy of our approach is most pronounced on ambiguity-sensitive models like DIN-SQL and DAIL-SQL. Table 4 quantifies this by introducing the **recovery rate**the percentage of the performance gap closed by refinement. These models recover a substantial portion of their initial performance loss, with recovery rates reaching up to 75% on the development set, thereby attaining over 92% of their original "clean" accuracy. In contrast, the change for MAC-SQL is minimal, a result consistent with the higher robustness to ambiguity it already demonstrated in RQ1. These findings establish that upstream schema refinement is a highly effective and practical strategy for restoring the accuracy of downstream Text-to-SQL models.

Table 4: RQ3: Percentage of performance *recovered* relative to clean Spider and percentage of *attained* clean accuracy after refinement. Recovery = (Refined − Dirty)/(Clean − Dirty) × 100%. Attained = Refined/Clean × 100%.

| Model | Dev (%) | | Test (%) | |
|---|---|---|---|---|
| | Recovery | Attained | Recovery | Attained |
| DIN-SQL | 74.42 | 92.44 | 52.63 | 92.02 |
| DAIL-SQL | 75.62 | 92.98 | 44.76 | 83.53 |
| MAC-SQL | 6.28 | 92.83 | 2.78 | 95.75 |

### 4.6 ANALYSIS OF RESIDUAL ERRORS

While schema refinement improves downstream accuracy, the results indicate that a gap to the original "clean" performance persists (see "Attained" column, Table 4). An analysis of these failure cases suggests they can be attributed to scenarios where the semantic content of the database itself is insufficient for correct disambiguation. We categorize these scenarios into two primary types:

**Low-Signal Data Content.** In certain instances, the data values within a column provide insufficient semantic information for disambiguation. For example, in the `cre_Doc_Template_Mgt` database, a column's probed values consist of single characters such as `[n, y, u, h]`. This low-signal content does not allow the model to distinguish between the concepts of a "description" and a "code." As a result, the refinement algorithm outputs `document_code`, a plausible but incorrect name relative to the ground truth (`Document_Description`). This discrepancy leads to query failures on questions targeting the document's description.

**Mixed-Semantic Data Content.** In other instances, the set of probed data values contains multiple distinct semantic types. For example, a column in the `employee_hire_evaluation` database contains values `[Bristol, Bath, Wasps, Sale]`, a mixture of city names and sports team names. This compositional ambiguity in the input data results in the refiner algorithm selecting `employee_team_name` as the canonical name, which diverges from the ground-truth concept of `city`. Consequently, questions about employee city are answered incorrectly.

**Boundary Conditions.** These failure cases illustrate a boundary condition of the proposed method: its performance is dependent on the semantic clarity of the information available within the database schema and its contents. When the input data itself exhibits low semantic signal or high semantic ambiguity, the refinement algorithm may generate an output that, while consistent with the immediate evidence, is globally incorrect. Addressing these cases may require mechanisms beyond local context analysis, such as the integration of external knowledge or human-in-the-loop validation, representing potential directions for future research.

## 5 CONCLUSION

This work systematically investigates the impact of lexical-schema ambiguity on Text-to-SQL systems and evaluates the efficacy of an automated, upstream refinement strategy. Our experiments first quantified the performance degradation caused by such ambiguity, establishing a clear need for remediation. We then introduced SCHEMA-REFINER, a method that leverages graph-based context extraction, and demonstrated its ability to accurately identify different types of ambiguity. Crucially, we showed that by generating queries against the refined schema, a substantial portion of the initial performance loss can be recovered on downstream execution accuracy, particularly for ambiguity-sensitive models. Our analysis also identified key boundary conditions, showing that the refinement process is constrained by the semantic quality of the underlying data content. Taken together, these findings establish upstream schema refinement as a viable and effective strategy for improving the robustness of Text-to-SQL systems in environments with imperfect metadata, and frame data-level semantic clarity as a critical factor for future research.

## 6 ETHICS STATEMENT

This work complies with the ICLR Code of Ethics and conference policies; it uses only publicly available data, involves no human subjects or personally identifiable information, and adheres to legal, privacy, fairness, and research-integrity standards throughout.

## 7 REPRODUCIBILITY STATEMENT

All datasets referenced in this paper and the code for our experiments (including configuration files and evaluation scripts) are provided in the anonymized supplementary material to facilitate independent reproduction of our results.

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

## A    LLM USAGE

Large Language Models (LLMs) were used to aid in the writing and polishing of the manuscript. Specifically, we used an LLM to assist in refining the language, improving readability, and ensuring clarity in various sections of the paper. The model helped with tasks such as sentence rephrasing, grammar checking, and enhancing the overall flow of the text.

It is important to note that the LLM was not involved in the ideation, research methodology, or experimental design. All research concepts, ideas, and analyses were developed and conducted by the authors. The contributions of the LLM were solely focused on improving the linguistic quality of the paper, with no involvement in the scientific content or data analysis.

The authors take full responsibility for the content of the manuscript, including any text generated or polished by the LLM. We have ensured that the LLM-generated text adheres to ethical guidelines and does not contribute to plagiarism or scientific misconduct.

