# OpenReview forum: "Schema-Refiner: Synergizing Knowledge Graphs and LLMs for Proactive Schema Refinement in Text-to-SQL"
_ICLR.cc/2026/Conference — ICLR 2026 Conference Withdrawn Submission_

### Official Review · Reviewer_wJE5 · 2025-10-21

**Soundness:** 2
**Presentation:** 1
**Contribution:** 2
**Rating:** 2
**Confidence:** 5

**Summary:**

The paper attempts to address the challenge of linking text to schema elements in the Text-to-SQL task.  They focus on hononymy (different columns in the schema with similar names), synonymy, and uninformative column names.  They propose a pipeline based on first infering a column context based on column-values, and other columns of the schema represented as a knowledge graphs, getting canonical names of the columns using an LLM, and then based on some logic creating views in terms of informative, un-ambigous column names.  Queries are generated on the view, and programmatically transformed to SQL on the actual physical schema.

**Strengths:**

* The idea of creating a virtual clean schema as a view makes a lot of sense in practice.

* The experimental results show significant gains with their proposed strategy of creating virtual schema

* The problem tackled in the paper is of great practical importance.

**Weaknesses:**

* W1 Important details needed to assess the worth of their proposed claims are missing.  The paper mostly sketches out the top-level idea, but does not explain the working of each step of their idea.
     * W1(a)  A crucial missing part is on the creation of views.  After the tests in Equations (6)--(8) are conducted, on what set are the views created and how?
     * W1 (b) To which knowledge graph is the schema mapped?
     * W1 (c) How eactly is the community discovery algorithm run? The paper mentions a proper noun (Leiden) as a the community discovery algorithm without any citation.

W2 The paper defines homonymy etc purely in terms of column names. However, an absolute column name does not make sense.  Shouldn't all your definitions prefix each column name with the name of its containing table.  Generic column names like "name" and "id" are often repeated across tables in a schema, and they are not to be treated stand-alone.

W3 The experiments are not reproducible.  Paper does not provide details of how exactly they obtained the ambiguous schema starting from the Spider schema.

W4 The topic of ambiguous or noisy column names has been studied before, and the paper cites those papers, however their discussion of the related work is perfunctionary and does not present nuanced analysis of prior work and technically how their approach differs.  Related work is too generic and high-level and reads like it is LLM-generated.

W5  The paper crucially relies on the LLM to get canonical schema names based on column-context and sampled values.  How will this help for numerical columns or with private niche domain specific schema?  Also, please note that almost all SOTA Text-to-SQL systems attach sampled values with column names when inputing the schema to the LLM for text-to-sql generation.

W6 The paper does not consider SOTA text-to-sql systems in its evaluations.

W7 The paper does not empircally compare with any of the prior work on mitigating effect of ambiguous schema.

**Questions:**

Please address the concerns raised in the weak points above.  Other minor concerns below.

In line 187, the set of candidate column pairs for synonymy detction seems needless large.

On what basis are the three thresholds chosen? Would these work for schema where table names differ from each other only by a small suffix (example date range).  Please see the Spider V2 dataset schema.

In Figure 1, it is not at all clear how LLM helps identify that the salaries are monthly, not annual.  How is the full name extracted from the given schema?

---

### Official Review · Reviewer_wutB · 2025-10-29

**Soundness:** 2
**Presentation:** 2
**Contribution:** 1
**Rating:** 2
**Confidence:** 4

**Summary:**

The paper addresses lexical-schema ambiguity (homonymy, synonymy, irregular naming) in Text-to-SQL by building a schema knowledge graph, using Leiden community detection to extract context, employing LLMs to infer canonical names, and generating CREATE VIEW statements for a clean logical layer. They introduce Amb-Spider, a benchmark with controlled ambiguity injection, and show that state-of-the-art Text-to-SQL models degrade 3-25% on ambiguous schemas but recover 45-75% of lost performance with their refinement.

**Strengths:**

1. Schema ambiguity does hurt Text-to-SQL systems. The 20+ percentage point drops on DIN-SQL/DAIL-SQL validate this matters.
2. CREATE VIEW layer is pragmatic—no database modification, queries auto-rewritten back to original schema.

**Weaknesses:**

1. Synthetic benchmark when real ones exist: Amb-Spider artificially injects ambiguities into the clean Spider dataset, but Spider 2.0 and BIRD already contain real enterprise-level schema messiness. Testing on synthetic data doesn't validate whether the approach works on actual production databases. The paper should evaluate on Spider 2.0/BIRD to demonstrate real-world effectiveness.

2. Insufficient literature review: Schema graphs have been standard in Text-to-SQL (RAT-SQL, BRIDGE, LGESQL). The graph construction is similar to RAT-SQL. The paper needs to properly situate contributions within existing work and clearly articulate what's fundamentally new beyond applying 2024 LLMs to established techniques.

3. Outdated technical design: The pipeline uses Leiden (2019 structural algorithm) when modern LLM embeddings capture both structure and semantics. More critically, the system uses embeddings + threshold grid search (testing 64 threshold combinations) for classification when direct LLM prompting ("are these columns the same concept?") would be simpler, require no tuning, and provide better contextual reasoning at comparable cost.

**Questions:**

1. Can you provide results on Spider 2.0 or BIRD to validate the approach works on actual enterprise databases rather than synthetic ambiguity?

2. Direct LLM classification: What's the performance and cost comparison between your embedding + threshold approach versus directly asking GPT-4 to classify ambiguities? Include ablation study showing whether the added complexity provides meaningful benefit.

3. Documentation baseline: For databases with existing metadata (even if incomplete), how does Schema-Refiner compare to simply providing that documentation to the Text-to-SQL model? What percentage of real databases lack sufficient documentation to make your approach necessary?

---

### Official Review · Reviewer_WNY2 · 2025-11-03

**Soundness:** 2
**Presentation:** 2
**Contribution:** 1
**Rating:** 2
**Confidence:** 4

**Summary:**

This paper tackles the problem of lexical–schema ambiguity in Text-to-SQL tasks, where ambiguous or inconsistent column/table names lead to reduced model accuracy. The authors propose Schema-Refiner, a neuro-symbolic pipeline that proactively refines database schemas before query generation. It builds a schema knowledge graph, applies community detection, uses an LLM to infer canonical names, and generates CREATE VIEW statements to create a cleaner logical schema layer. A new benchmark, Amb-Spider, is introduced by injecting ambiguity into Spider to test this setting. Experiments show that ambiguity significantly degrades Text-to-SQL performance and that Schema-Refiner partially recovers the lost accuracy.

**Strengths:**

- Clear motivation and problem importance. Lexical–schema ambiguity is a real and underexplored issue for Text-to-SQL systems, and the paper articulates this motivation clearly.
- Well-structured and interpretable system design.
- The pipeline is easy to follow and combines knowledge graphs, community detection, and LLM-based canonicalization into a cohesive workflow. The non-destructive CREATE VIEW strategy is practical.

**Weaknesses:**

- Limited novelty and technical depth.
While the integration of different components is thoughtful, the method itself feels incremental. It primarily combines existing techniques—graph construction, LLM inference, and view generation—without introducing new learning formulations or algorithms. The contribution is mainly system-engineering oriented rather than algorithmic, which falls short of ICLR’s emphasis on methodological innovation.
- Narrow and synthetic evaluation.
The evaluation focuses on demonstrating that ambiguity harms Text-to-SQL and that Schema-Refiner can mitigate this harm. However, there is no comparison to current state-of-the-art models under realistic or noisy conditions, and the benchmark (Amb-Spider) is synthetically generated by LLMs rather than derived from real-world databases. As a result, the practical and external validity of the findings is limited.
- Generalizability and scalability concerns.
The pipeline appears ad-hoc and domain-specific, relying on multiple hand-tuned thresholds and prompting heuristics. Its performance on large-scale, multilingual, or cross-domain schemas is unclear. The dependence on repeated LLM inference also raises scalability and cost concerns.
- No analysis of efficiency or deployment overhead.
Schema refinement involves multiple LLM calls and graph operations, but the paper does not provide runtime or computational cost analysis. The absence of efficiency metrics weakens the practical argument for adoption.

**Questions:**

The paper presents a meaningful exploration of schema ambiguity and a well-motivated system design. However, its novelty and generalizability are limited, and the evaluation does not convincingly establish that Schema-Refiner represents a substantive advance over existing approaches.

---

### Official Review · Reviewer_bVhx · 2025-11-03

**Soundness:** 2
**Presentation:** 3
**Contribution:** 2
**Rating:** 2
**Confidence:** 5

**Summary:**

The paper focuses on lexical–schema ambiguity in Text-to-SQL systems, which arises from homonymy, synonymy, and irregular table/column naming and degrades accuracy in real-world databases. The authors propose Schema-Refiner, a neuro-symbolic framework that builds a schema-based knowledge graph, performs community detection, queries an LLM to infer canonical field names and semantics, and automatically generates CREATE VIEW statements to expose a clean schema layer for downstream NL2SQL models. They also create Amb-Spider, an injected-ambiguity benchmark derived from Spider. Experiments show large accuracy drops under ambiguity and partial recovery when using the proposed refinement layer.

**Strengths:**

S1. The paper tries to address a real challenge, schema ambiguity, in Text-to-SQL. This reflects the real-life scenarios in many enterprise or open databases.

S2. The view creation essentially builds a semantic layer on top of the ambiguous schema, which provides standard/clean contextual information for Text-to-SQL.

S3. The newly created benchmark demonstrates non-trivial performance degradation with introduced noise.

**Weaknesses:**

W1. The paper has limited novelty. The proposed solution essentially combines a set of off-the-shelf components (KG, LLM prompting, and view-based abstraction). As a result, the contribution reads more as a system integration effort.

W2. Synthetic ambiguity injection may not reflect realistic enterprise schema drift. It is unclear whether Amb-Spider aligns with real production workloads. Moreover, the approach has not been evaluated on large/complex enterprise schemas with thousands of tables/columns, where schema noise, denormalization, and missing relational structure (pk/fk) are common, raising concerns about scalability and practical applicability.

W3. The proposed method heavily relies on LLM heuristics with insufficient ablations to isolation the value of each component (e.g., prompt design, KG integration vs. table context alone). Key design choices such as similarity thresholds and ambiguity definitions feel heuristic rather than grounded in principled semantic criteria.

W4. The synthetic dataset could be overfitting to Spider training distributions via semantics learned from dataset itself. Moreover, Spider dataset is relatively  small and structurally simple compared to enterprise database settings. The evaluation focuses primarily on execution accuracy.

**Questions:**

Q1. How realistic is Amb-Spider compared to real corporate schema? Any human-in-the-loop/SME validation?

Q2. How often do canonical names diverge from true business semantics even on clean Spider?

Q3. Did you evaluate simple baselines (e.g., camelCase → words, prefix/suffix normalization, column-embedding clustering)? For example, NameGuess introduce a suite of techniques to introduce noise to database schema.

Q4. What is the computational cost? LLM inference for every column can be expensive.

Q5. Can the system gracefully handle mixed semantic fields or JSON column payloads in modern warehouses?

Q6. Does this generalize to enterprise schemas with thousands of tables, weak foreign key use, or buried semantic constraints?

Q7. Is data probing feasible in regulated or realistic settings where direct data sampling is restricted or the query workload is under control?

---

### Note · Authors · 2025-11-12

I have read and agree with the venue's withdrawal policy on behalf of myself and my co-authors.